# Reducing the Adverse Effects of Salt Stress by Utilizing Compost Tea and Effective Microorganisms to Enhance the Growth and Yield of Wheat (*Triticum aestivum* L.) Plants

Mohssen Elbagory [1,2]

1    Department of Biology, Faculty of Science and Arts, King Khalid University, Mohail 61321, Saudi Arabia; mhmohammad@kku.edu.sa

2    Agricultural Research Center, Department of Microbiology, Soils, Water and Environment Research Institute, Giza 12112, Egypt

**Abstract:** One of the worst environmental conditions limiting crop plant productivity is salinity. As a result, ecologically friendly methods are urgently needed to boost the development and yield of wheat growing on saltine soils. Two-year field studies to examine the effects of applying compost tea (CT) and effective microorganisms (EMs; *Azospirillum brasilense*, *Pseudomonas koreensis*, and *Bacillus circulans*) on the growth and yield of two wheat cultivars, namely Sids 12 as a salinity susceptible cultivar and Misr 1 as a salinity resistant cultivar, under salt-affected soils. The findings corroborated our hypothesis that, in comparison to the control and the individual applications of EM or CT, the combined application (EM + CT) significantly improved growth, yield, uptake of nutrients, and photosynthetic characteristics. Furthermore, the combined application markedly ($p \leq 0.05$) boosted the antioxidant enzymes. Our research showed that the combination treatment could increase soil microbial activity and activate critical soil enzymes, primarily dehydrogenase and urease. In general, the combination treatment has demonstrated a good effect in terms of stimulating plant development and raising element concentrations in wheat under salt stress.

**Keywords:** wheat; salinity stress; beneficial microorganisms; foliar application; growth dynamics; productivity

## 1. Introduction

Wheat production is under significant pressure from the rapidly expanding population, and by 2050, consumer demand will be 60% higher than it is today [1]. In terms of basic needs (food, feed, and biofuel security), wheat is one of the most strategic crops in the world [2]. In Egypt, there were 1,394,558 hectares of wheat fields in cultivation in 2021, yielding 9.00 Mt [3]. Wheat productivity can be increased by better managing soil and water under salt-affected soils. Cultivated soil all over the world has become saltier (ESP < 15%; EC > 4 and pH: 7.5–8.5), due to insufficient irrigation water, desertification processes, and fertilization, and >800 Mha of soil are affected by salt stress globally [4]. In parallel, excessive concentrations of $Na^+$ and $Cl^-$, lower soil water potential, nutritional imbalance, and salt stress all affect plant development and production [5]. For the aforementioned reasons, increasing yield is a crucial area of our research. There are numerous methods—many of which are economical and environmentally friendly—to reduce toxic effects by using efficient microorganisms (EMs) and organic nutrients, such as compost tea (CT), which are the most promising methods for fostering soil health and crop growth dynamics [6,7]. Several EM genera, i.e., *Pseudomonas*, *Bacillus*, *Azospirillum,* and *Bradyrhizobium*, include a few species that have been found to enhance the growth and production of several crops planted in salinity-affected soils [6]. A high $K^+/Na^+$ ratio, production hormones (auxins, cytokinins, and gibberellins), the scavenging of reactive oxygen species (ROS), the stimulation of soil enzyme activity, and the improvement of various soil properties can all result in an

increase in the germination of seeds treated with EMs [8–10]. Additionally, foliar spraying on plants is a successful strategy that has replaced other agricultural technologies used to decrease the effects of salt stress and regulate growth [11]. One of the most important of these alternative methods is the foliar application of compost tea, which riches with humic acids, hormones, amino acids, vitamins, minerals, and beneficial microbes. CT can improve the growth and yield of crops and their resistance to disease [12–14]. As a foliar spray or root drench, CT can be used to promote the elongation of roots and growth by producing cytokinins and gibberellic acids, as well as to buffer soil pH by containing organic acids and humic compounds [15–18]. Studies have demonstrated that CT has a positive effect on growth, quality, and production while also enhancing the activities of enzymatic antioxidants [14,19].

This study assessed the effects of combining EM (*Azospirillum brasilense, Pseudomonas koreensis*, and *Bacillus circulans*) and CT on different growth parameters, namely vegetative, physiological, nutrient content, soil bioactivity, and production, for two cultivars of wheat plants, Sids 12 and Misr 1, in order to mitigate the detrimental impacts of salt-affected soils.

## 2. Materials and Methods

### 2.1. Description of the Site

Under salt-affected soil conditions (EC, 10.04–10.23 dSm$^{-1}$), effective microorganisms and foliar spray with compost tea (CT) treatments were applied to enhance the growth and yield of 2 cultivars of wheat (Sids 12 as a susceptible cultivar and Misr 1 as a resistance cultivar). Two field trials (winter of 2019–2020 and winter of 2020–2021), were conducted at the North Delta of Egypt, Sakha Agricultural Research Station Farm. The average values for meteorological data during the growing seasons were recorded as 24.01 °C, 13.25 °C and 2.15 mm in the 1st season and as 22.8 °C, 12.86 °C and 2.18 mm in the 2nd season for maximum, minimum temperature and rainfall, respectively. Table 1 displays some different properties of the field soil during the two growing seasons.

**Table 1.** Some physicochemical and biological properties of soil used in the two winter growing seasons.

| Season | Feature | Value |
|---|---|---|
| 2019/2020 | Texture | Clayey (USDA) |
| | pH (1: 2.5) | 8.49 |
| | EC | 10.04 dSm$^{-1}$ |
| | OM | 15.33 g kg$^{-1}$ |
| | Nitrogen | 16.20 mg kg$^{-1}$ |
| | Phosphorus | 10.76 mg kg$^{-1}$ |
| | Potassium | 365.00 mg kg$^{-1}$ |
| | Counts of bacteria (CFU $\times$ 10$^6$ g$^{-1}$ dry soil) | 127 |
| 2020/2021 | Texture | Clayey (USDA) |
| | pH (1: 2.5) | 8.27 |
| | EC | 10.23 dSm$^{-1}$ |
| | OM | 17.10 g kg$^{-1}$ |
| | Nitrogen | 14.33 mg kg$^{-1}$ |
| | Phosphorus | 9.91 mg kg$^{-1}$ |
| | Potassium | 389.12 mg kg$^{-1}$ |
| | Counts of bacteria (CFU $\times$ 10$^6$ g$^{-1}$ dry soil) | 178 |

EC: Electrical conductivity; OM: Organic matter; Nitrogen, Phosphorus, and Potassium as available.

### 2.2. Effective Microorganisms (EMs)

From the Microbiology Department, SWERI, ARC, Egypt, different bacterial strains: *Azospirillum brasiliense* SARS 1001, *Pseudomonas koreensis* MG209738, and *Bacillus circulans* NCAIM B.02324, were provided. Cultures of *Azospirillum, Pseudomonas*, and *Bacillus* strains were maintained on Semi-Solid Malate media [20], King's B (KB) broth medium [21], and Nutrient Broth medium [22], respectively. During the sowing of wheat seeds, a mix of the

investigated bacterial strains (1:1:1, with a rate of 1.4 Kg ha$^{-1}$) was generated as peat-based inoculums.

### 2.3. Compost Tea (CT)

During the growth stage (30 and 50 days), CT (140 L ha$^{-1}$) was applied as a foliar treatment using a hand atomizer. The CT had the following chemical and biological characteristics: pH, 6.7 and 6.5; EC, 2.77 and 2.98 dS m$^{-1}$; total N, 5120 and 5280 ppm; available P, 3349 and 3709 ppm; available K, 4344 and 4512 ppm; total count of bacteria (Log CFU ml$^{-1}$), 7.11 and 7.45; total count of actinomycetes (Log CFU ml$^{-1}$), 4.55 and 4.98; total count of fungi (Log CFU ml$^{-1}$), 4.21 and 3.98, during the two growth seasons 2019/2020 and 2020/2021, respectively. CT was provided by the biofertilizers production unit, SWERI, Kafr El-Sheikh.

### 2.4. Field Trials

The trials (4 m $\times$ 4 m/plot) were conducted in triplicates using a split–split plot design. The two growing seasons, 2019/2020 and 2020/2021, were taken into consideration as the main plot. Two wheat cultivars (Sids 12 as a salinity susceptible cultivar and Misr 1 as a salinity-resistant cultivar) from the Sakha Agricultural Research Station were used for the subplots, while the sub–sub plots treatment were the control, EM, CT, and both EM + CT. On November 10, during the 1st season, and on November 14, during the 2nd season, wheat seeds were drilled with 140 kg ha$^{-1}$.

Calcium superphosphate as P fertilizer (15.5% $P_2O_5$) and potassium sulfate as K fertilizer (48% $K_2O$) were given to the soil during ploughing at rates of 360 and 120 kg ha$^{-1}$, respectively. Parallel to these treatments, urea as N fertilizer (46.5% N) was applied in two equal doses (within 45 days of sowing), at a rate of 2/3 of the full dose (240 kg ha$^{-1}$) for EM treatments and the full dose for all other treatments (360 kg ha$^{-1}$). Four irrigations were used during the plant growth period at a rate of up to 400 m$^3$/irrigation.

### 2.5. Measurements

At 70 days after sowing, 6 random plants from each replication were chosen to measure various parameters, including vegetative growth, physiological changes, antioxidant enzymes, nutrient content, and soil enzymes. The yield (ton ha$^{-1}$) including grain, straw, and biological as well as the harvest index (%) was measured at 150 days. Measurement tape was used to determine the plant's height and the length of its roots (cm plant$^{-1}$), and an electronic scale was used to determine the plant's dry weight (ADAM, PW 214, UK). A UV spectrophotometer (Jenway 6705, UK) was used to measure total chlorophyll as mg g$^{-1}$ FW, carotenoids as μg g$^{-1}$ FW according to [23], proline as μmol g$^{-1}$ FW according to [24], and total soluble sugars (μg g$^{-1}$ FW) according to [25].

For the activity of antioxidant enzymes, catalase (CAT) activity was measured at 240 nm according to the techniques of [26], whilst ascorbate peroxidase (APX) activity was assessed at 290 nm according to the techniques described in [27]. Enzyme activities were expressed as μM $H_2O_2$ min$^{-1}$ g$^{-1}$ FW. In addition, according to [28], a Flame photometer was used to measure $K^+$, $Na^+$, and $k^+/Na^+$.

In the rhizosphere, dehydrogenase (DAH) activity was estimated at 485 nm (mg TPF g$^{-1}$ soil day$^{-1}$) by [29], whereas urease activity was estimated at 410 nm (mg $NH_4^+$–N g$^{-1}$ soil day$^{-1}$). At a standard moisture level of 14%, the plot was harvested, and the yield was weighed and converted to hectares. In addition, the harvest index (HI), was calculated and represented in %.

### 2.6. Statistical Analysis

The data were statistically analyzed using Co Stat's statistics software (6.303). The various treatments were compared using an ANOVA, and according to [30], multiple comparisons at $p \leq 0.05$ were completed using Duncan's range tests.

## 3. Results

### 3.1. Vegetative Parameters

The findings demonstrated that the application of EM, CT, and the mixture led to significant impacts ($p \leq 0.05$) on the vegetative parameters of different cultivars of wheat plants (Sids 12 and Misr 1) under salt-affected soil conditions in the 1st and 2nd seasons (Figure 1). In general, EM + CT boosted the growth parameters compared to the control treatment. Seventy days after sowing in season 2019/2020, plants treated with a soil and foliar application had significantly increased plant height (cm plant$^{-1}$) for Sids 12 and Misr 1 from 63.12 and 69.58 (control, T1) to 79.63 and 77.70 (combination, T4), whereas the same treatment increased root length (cm plant$^{-1}$) for Sids 12 and Misr 1 from 18.17 and 19.80 (control, T1) to 22.73 and 22.15 (Figure 1), respectively. The second season (2020–2021) showed a similar trend. The negative results of salt stress on the dry weight of the studied wheat cultivars were also lessened with soil and foliar application. For Sids 12 and Misr 1, the T4 treatment was more effective, yielding 3.98 and 3.89 g plant$^{-1}$ in the 2019–2020 season and 4.06 and 3.94 g plant$^{-1}$ in the 2020–2021 season (Figure 1), respectively.

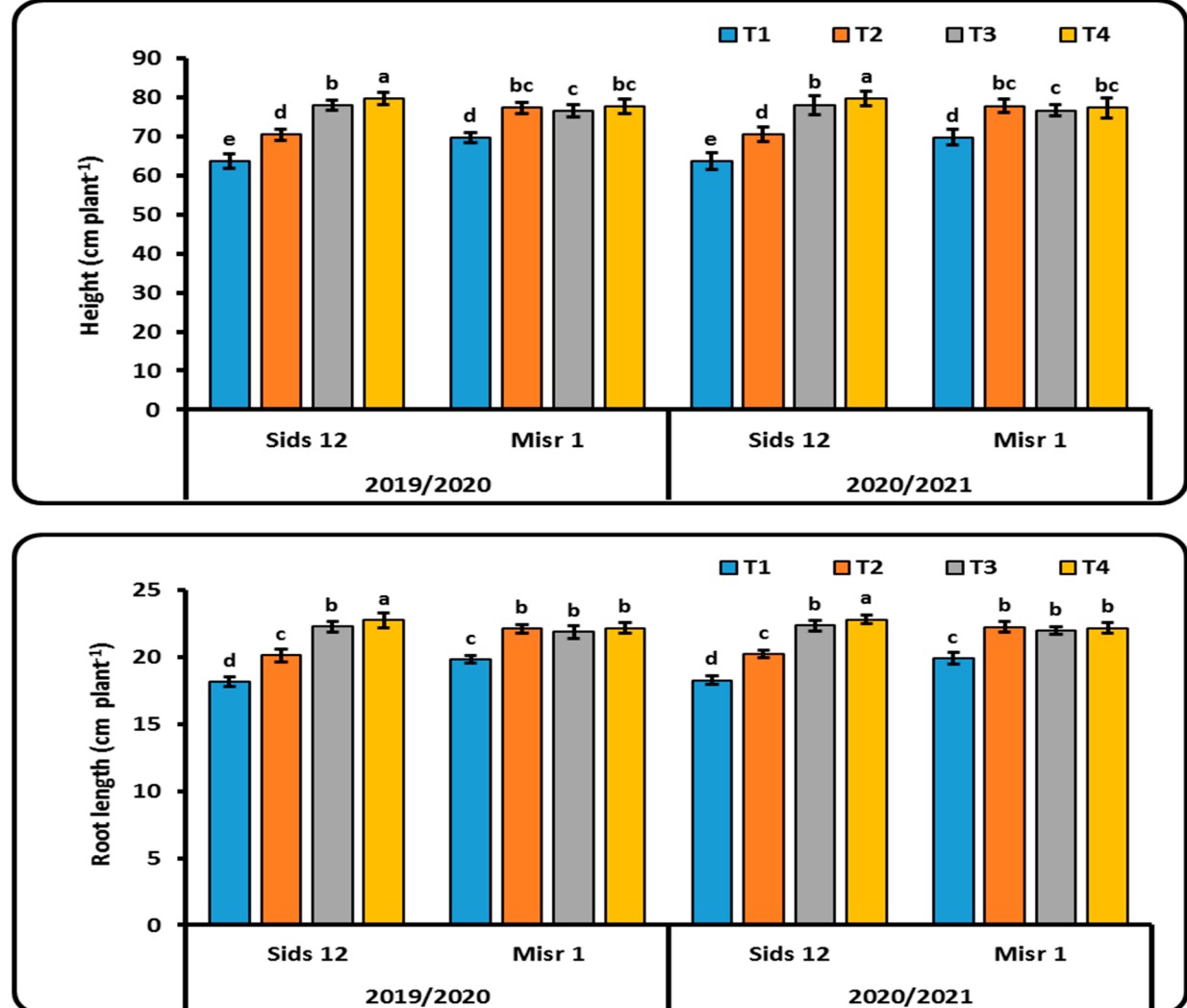

**Figure 1.** *Cont.*

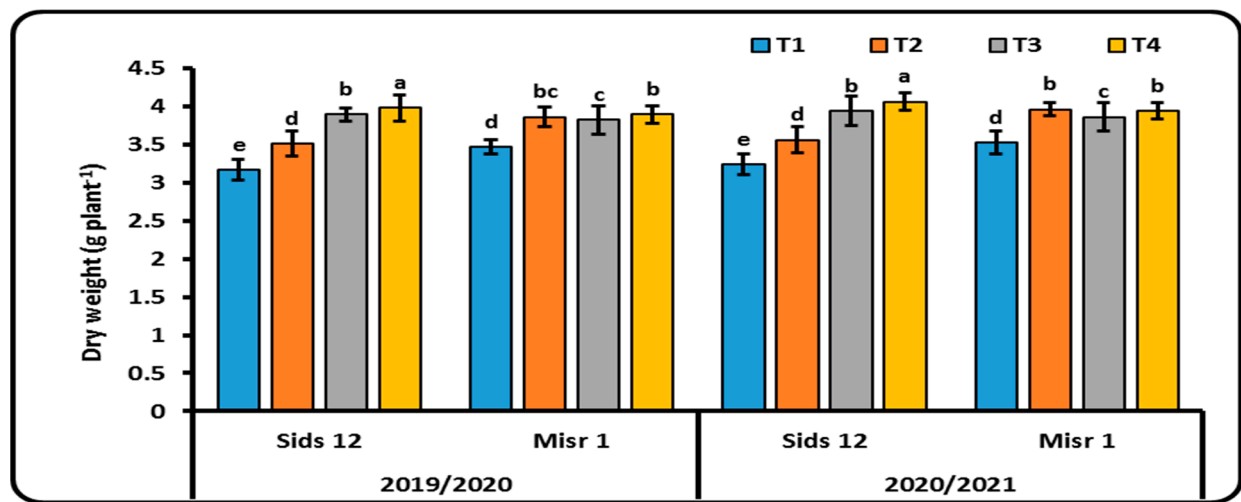

**Figure 1.** Impact of different treatments (EM, CT, and combination) on some vegetative growth parameters 70 days after sowing during the two growing seasons. Duncan's test ($p \leq 0.05$); means for the same growing season denoted with different letters indicate significant differences across treatments. ($\pm$) means standard deviations (SD, n = 3). Control (T1); EM (T2); CT (T3); EM + CT (T4).

*3.2. Physiological Parameters*

Seventy days after sowing, some physiological parameters of the studied wheat cultivars exhibited significant variations with respect to the tested treatments, T1 (control), T2 (EM), T3 (CT), and T4 (EM + CT) (Table 2). With regard to the growing season, the second season recorded high values in physiological traits compared to the first season, while the Misr 1 cultivar scored the highest values when compared to the Sids 12 cultivar. For the studied treatments, the combined treatment (EM + CT) recorded the greatest values of 3.32 for total chlorophyll (mg g$^{-1}$ FW), 1.12 for carotenoids (µg g$^{-1}$ FW), 7.97 for proline (µmol g$^{-1}$ FW), and 4.99 for TSS (µg g$^{-1}$ FW) compared to control and other treatments. On the other hand, there were no significant differences in cultivar x Year (Table 2).

**Table 2.** Impact of different treatments (EM, CT, and combination) on physiological characteristics of wheat plants at 70 days after sowing, during the two growing seasons.

| Source of Variation | Total Chlorophyll (mg g$^{-1}$ FW) | Carotenoids (µg g$^{-1}$ FW) | Proline (µmol g$^{-1}$ FW) | Total Soluble Sugars (µg g$^{-1}$ FW) |
|---|---|---|---|---|
| Year | | | | |
| 2019/2020 | 3.11 ± 0.51 [b] | 0.91 ± 0.19 [b] | 7.07 ± 1.11 [b] | 4.05 ± 0.41 [b] |
| 2020/2021 | 3.26 ± 0.62 [a] | 1.06 ± 0.11 [a] | 7.29 ± 1.23 [a] | 4.20 ± 0.54 [a] |
| Significance | ** | ** | ** | ** |
| Cultivar | | | | |
| Sids 12 | 3.13 ± 0.19 [b] | 0.93 ± 0.14 [b] | 7.07 ± 0.97 [b] | 3.64 ± 0.64 [b] |
| Misr 1 | 3.23 ± 0.13 [a] | 1.03 ± 0.15 [a] | 7.22 ± 1.02 [a] | 4.61 ± 0.81 [a] |
| Significance | ** | ** | ** | ** |
| Treatments | | | | |
| T1 | 3.06 ± 0.17 [c] | 0.85 ± 0.15 [c] | 6.61 ± 1.23 [d] | 3.39 ± 0.47 [d] |
| T2 | 3.06 ± 0.19 [c] | 0.86 ± 0.17 [c] | 7.27 ± 1.14 [b] | 4.21 ± 0.51 [c] |
| T3 | 3.28 ± 0.12 [b] | 1.08 ± 0.16 [b] | 6.73 ± 0.99 [c] | 4.41 ± 0.49 [b] |
| T4 | 3.32 ± 0.14 [a] | 1.12 ± 0.21 [a] | 7.97 ± 1.12 [a] | 4.99 ± 0.52 [a] |
| Significance | ** | ** | ** | ** |
| Interaction | | | | |
| Cultivar X Treatment | ** | ** | ** | ** |
| Treatment X Year | NS | NS | NS | ** |
| Cultivar X Year | NS | NS | NS | NS |

Duncan's test ($p \leq 0.05$); means for the same growing season denoted with different letters indicate significant differences across treatments. ($\pm$) means standard deviations (SD, n = 3). Control (T1); EM (T2); CT (T3); EM + CT (T4). NS: not significant; **: Highly significant.

### 3.3. Antioxidant Enzymes

The results shown in Table 3 indicate that different treatments with EM, CT, and their mixture over the two growing seasons significantly affected the activity of antioxidant enzymes in the examined cultivars of wheat plants (Sids 12 and Misr 1). Different treatments considerably improved catalase and ascorbate peroxidase activity in wheat leaves 70 days after sowing compared to the control (Table 3). High significance was observed in the different wheat cultivars for catalase and APX activity, with values of 19.93 and 377.66 ($\mu$M $H_2O_2$ $min^{-1}$ $g^{-1}$ FW) for Sids 12 and 20.40 and 391.58 ($\mu$M $H_2O_2$ $min^{-1}$ $g^{-1}$ FW) for the Misr 1 cultivar, respectively. With regard to different treatments, the foliar application treatment (CT, T3) significantly improved catalase and APX activity ($\mu$M $H_2O_2$ $min^{-1}$ $g^{-1}$ FW), which attained values of 23.54 and 419.67 compared to the control and other treatment (Table 3).

**Table 3.** Impact of different treatments (EM, CT, and combination) on antioxidant enzymes of wheat 70 days after sowing during the two growing seasons.

| Source of Variation | Catalase ($\mu$M $H_2O_2$ $min^{-1}$ $g^{-1}$ FW) | Ascorbate Peroxidase ($\mu$M $H_2O_2$ $min^{-1}$ $g^{-1}$ FW) |
|---|---|---|
| Year | | |
| 2019/2020 | 19.13 $\pm$ 1.21 [b] | 366.87 $\pm$ 26.23 [b] |
| 2020/2021 | 21.20 $\pm$ 1.34 [a] | 392.37 $\pm$ 32.12 [a] |
| Significance | ** | ** |
| Cultivar | | |
| Sids 12 | 19.93 $\pm$ 1.04 [a] | 377.66 $\pm$ 29.91 [a] |
| Misr 1 | 20.40 $\pm$ 1.11 [a] | 391.58 $\pm$ 27.23 [a] |
| Significance | NS | NS |
| Treatments | | |
| T1 | 17.42 $\pm$ 0.96 [d] | 328.00 $\pm$ 31.11 [c] |
| T2 | 18.23 $\pm$ 1.09 [c] | 354.00 $\pm$ 29.19 [b] |
| T3 | 23.54 $\pm$ 1.02 [a] | 419.67 $\pm$ 32.10 [a] |
| T4 | 21.47 $\pm$ 1.23 [b] | 416.83 $\pm$ 35.19 [a] |
| Significance | ** | ** |
| Interaction | | |
| Cultivar X Treatment | ** | ** |
| Treatment X Year | NS | NS |
| Cultivar X Year | NS | NS |

Duncan's test ($p \leq 0.05$); means for the same growth season denoted with different letters indicate significant differences across treatments. ($\pm$) means standard deviations (SD, n = 3). Control (T1); EM (T2); CT (T3); EM + CT (T4). NS: not significant; **: Highly significant.

### 3.4. Nutrient Percentage

In the investigated cultivars of leafy wheat plants, the combination treatment (EM + CT, T4) increased $K^+$% and the $K^+/Na^+$ ratio and decreased $Na^+$% with significant differences 70 days after planting under varied soil and foliar spray treatments (Table 4). The best treatment that produced a high percentage of $K^+$ and $K^+/Na^+$ was T4, which achieved increases of 3.12% and 1.39, respectively, compared to the T1 treatment (control), while producing the biggest decrease in $Na^+$%, going from 2.52% (Control, T1) to 2.25% (EM + CT, T4). Regarding the ratio of $K^+/Na^+$, the examined cultivars of wheat showed Misr 1 > Sids 12 and 1st > 2nd as the descending order of growing seasons (Table 4).

**Table 4.** Impact of different treatments (EM, CT, and combination) on $K^+$, $Na^+$, and the ratio of $K^+/Na^+$ of wheat 70 days after sowing during the two growing seasons.

| Source of Variation | $K^+$ (%) | $Na^+$ (%) | $K^+/Na^+$ Ratio |
|---|---|---|---|
| Year | | | |
| 2019/2020 | $2.91 \pm 0.27$ [b] | $2.30 \pm 0.16$ [b] | $1.27 \pm 0.12$ [a] |
| 2020/2021 | $3.06 \pm 0.33$ [a] | $2.46 \pm 0.15$ [a] | $1.25 \pm 0.15$ [b] |
| Significance | ** | ** | ** |
| Cultivar | | | |
| Sids 12 | $2.93 \pm 0.19$ [b] | $2.33 \pm 0.21$ [b] | $1.27 \pm 0.19$ [a] |
| Misr 1 | $3.03 \pm 0.25$ [a] | $2.42 \pm 0.25$ [a] | $1.24 \pm 0.15$ [b] |
| Significance | ** | ** | ** |
| Treatments | | | |
| T1 | $2.86 \pm 0.26$ [c] | $2.52 \pm 0.18$ [a] | $1.13 \pm 0.11$ [d] |
| T2 | $2.86 \pm 0.24$ [c] | $2.37 \pm 0.22$ [b] | $1.20 \pm 0.17$ [c] |
| T3 | $3.08 \pm 0.31$ [b] | $2.39 \pm 0.26$ [b] | $1.31 \pm 0.22$ [b] |
| T4 | $3.12 \pm 0.36$ [a] | $2.25 \pm 0.21$ [c] | $1.39 \pm 0.14$ [a] |
| Significance | ** | ** | ** |
| Interaction | | | |
| Cultivar X Treatment | ** | ** | ** |
| Treatment X Year | NS | ** | ** |
| Cultivar X Year | NS | NS | NS |

Duncan's test ($p \leq 0.05$); means for the same growth season denoted with different letters indicate significant differences across treatments. ($\pm$) means standard deviations (SD, n = 3). Control (T1); EM (T2); CT (T3); EM + CT (T4). NS: not significant; **: Highly significant.

### 3.5. Activities of Soil Enzymes

Dehydrogenase and urease activities in the soil were considerably higher in the wheat plant rhizosphere after combination treatment (EM + CT) 70 days after sowing than after the single treatment and control (Figure 2). For the Sids 12 and Misr 1 cultivars, respectively, the EM + CT treatment (T4) showed the maximum activity in DHA (mg TPF $g^{-1}$ soil day$^{-1}$) with values of 102.63 and 134.19, which was followed by the T2 treatment (EM) with values of 97.52 and 116.07. The second season showed the same pattern. The T4 treatment was the best treatment in terms of urease enzyme activity ($NH_4^+$–N $g^{-1}$ soil day$^{-1}$), recording values of 95.33 and 105.65 in the first growing season and 106.67 and 118.65 in the second growing season for the Sids 12 and Misr 1 cultivars, respectively, in comparison to the other treatments and control (Figure 2).

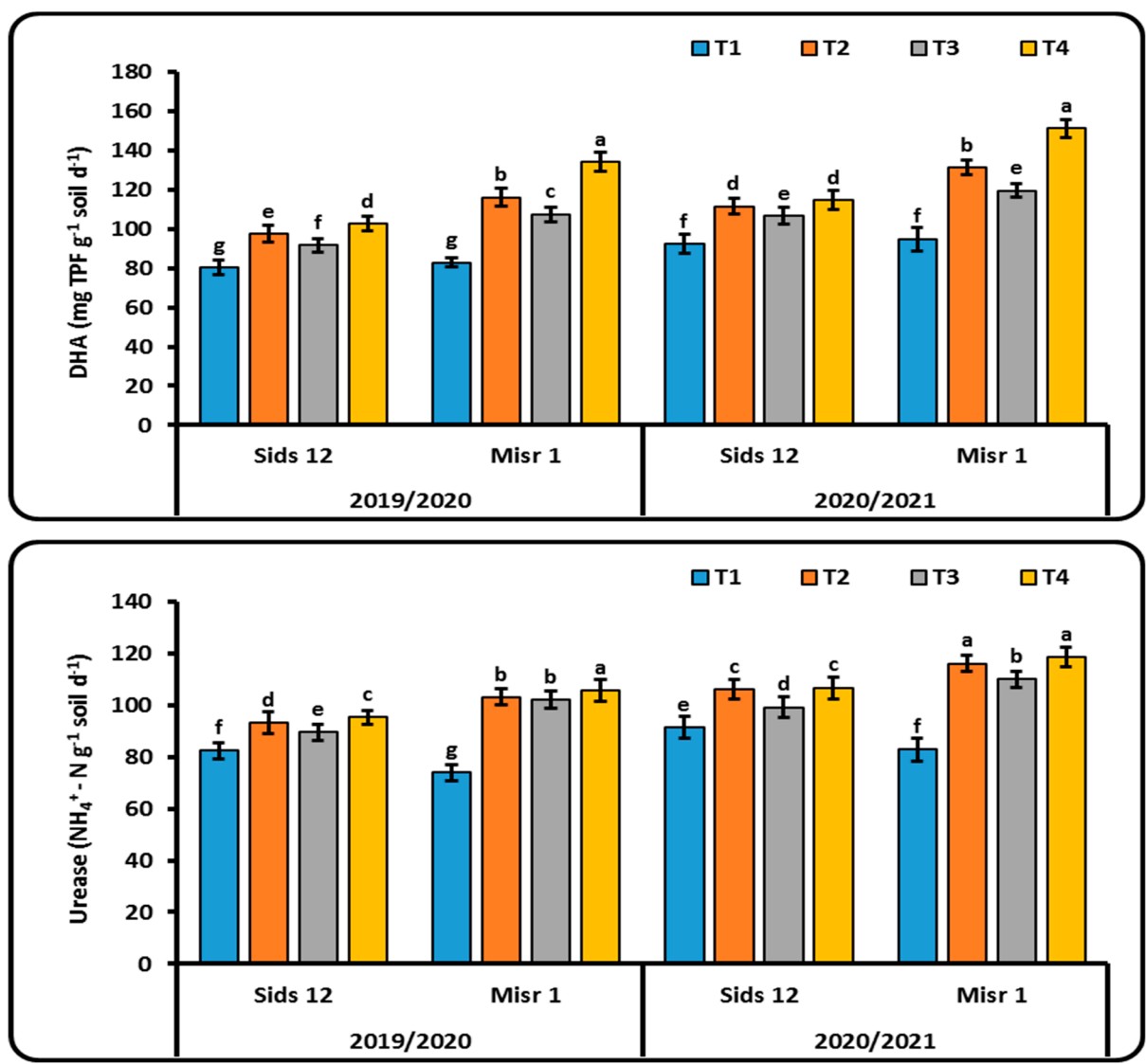

**Figure 2.** Impact of different treatments (EM, CT, and combination) on soil enzyme activities 70 days after sowing, during the two growing seasons. Duncan's test ($p \leq 0.05$); means for the same growth season denoted with different letters indicate significant differences across treatments. ($\pm$) means standard deviations (SD, n = 3). Control (T1); EM (T2); CT (T3); EM + CT (T4).

### 3.6. Yield

The results in Table 5 show that treatment (EM) in conjunction with CT during the two growing seasons considerably increased the yield and yield-related parameters of the two different cultivars of wheat plants. EM + CT (T4) caused the maximum values (ton ha$^{-1}$) of 7.98 and 12.13 for grain and straw yield compared to other treatments, respectively (Table 5). On the other hand, the Sids 12 cultivar recorded the lowest values of 16.95 and 39.64, compared to the Misr 1 cultivar, which recorded values of 17.66 and 39.92 for biological and harvest index (%), respectively.

According to the aforementioned data, Misr 1 > Sids 12 for the various wheat varieties, and T4 > T3 > T2 > T1 for soil and foliar applications.

**Table 5.** Impact of different treatments (EM, CT, and combination) on the yield and yield component of wheat 70 days after sowing during the two growing seasons.

| Source of Variation | Grain Yield (ton ha$^{-1}$) | Straw Yield (ton ha$^{-1}$) | Biological Yield | Harvest Index (%) |
|---|---|---|---|---|
| Year | | | | |
| 2019/2020 | 6.82 ± 0.39 [b] | 10.33 ± 1.12 [b] | 17.15 ± 1.34 [b] | 39.76 ± 1.45 [a] |
| 2020/2021 | 6.95 ± 0.42 [a] | 10.52 ± 1.08 [a] | 17.47 ± 1.27 [a] | 39.63 ± 1.67 [a] |
| Significance | ** | ** | ** | NS |
| Cultivar | | | | |
| Sids 12 | 6.72 ± 0.39 [b] | 10.23 ± 1.18 [b] | 16.95 ± 1.11 [b] | 39.64 ± 1.87 [b] |
| Misr 1 | 7.05 ± 0.41 [a] | 10.61 ± 1.19 [a] | 17.66 ± 1.23 [a] | 39.92 ± 1.97 [a] |
| Significance | ** | ** | ** | ** |
| Treatments | | | | |
| T1 | 5.17 ± 0.37 [d] | 9.51 ± 1.22 [d] | 14.68 ± 1.08 [d] | 35.21 ± 1.92 [d] |
| T2 | 6.98 ± 0.40 [c] | 9.78 ± 1.09 [c] | 16.76 ± 1.02 [c] | 41.64 ± 1.82 [b] |
| T3 | 7.41 ± 0.44 [b] | 10.27 ± 1.03 [b] | 17.68 ± 0.96 [b] | 41.91 ± 1.79 [a] |
| T4 | 7.98 ± 0.41 [a] | 12.13 ± 1.13 [a] | 20.11 ± 1.13 [a] | 39.68 ± 1.92 [c] |
| Significance | ** | ** | ** | ** |
| Interaction | | | | |
| Cultivar X Treatment | ** | ** | ** | ** |
| Treatment X Year | NS | NS | NS | NS |
| Cultivar X Year | NS | NS | NS | NS |

Duncan's test ($p \leq 0.05$); means for the same growing season denoted with different letters indicate significant differences across treatments. (±) means standard deviations (SD, n = 3). Control (T1); EM (T2); CT (T3); EM + CT (T4). NS: not significant; **: Highly significant.

## 4. Discussion

According to previous reports, the Mediterranean region's salt-affected soil has a detrimental effect on soil health and plant development, reducing plant production from 25 to 30% [31].

A significant abiotic stressor that also lowers agricultural soil production is salt-affected soil [32]. It is anticipated that the salinity of the soil will have a severe impact on the entire planet, wasting 30% of the land in the next 25 years and 50% in the next 50 years [33]. Wheat is sensitive to abiotic stresses because of the impact of osmotic stress on the development of the growth rate. The reduction in wheat growth rate under salt-affected soil stress is reported in the form of height and length of root as well as dry weight. The use of EM and CT promotes the growth of roots, which has an impact on the dynamics of plant growth. This advantageous result might be attributable to the function of EM and CT in nutrient- and water-transporting cell transport systems [34]. Additionally, the EM inoculation had a good impact on changes in the growth metrics. For instance, auxin, cytokinin, and gibberellin production are strongly associated to root morphogenesis and increased lateral root length and root hair density [19,35,36].

Previous research has demonstrated that plants (wheat) are more vulnerable to the negative effects of salinity, which results in a decrease in physiological and photosynthetic pigments. Therefore, the combination of microbial inoculation and foliar CT application improves a plant's photosynthetic system by increasing the number of leaves and their area, reducing osmotic stress in soils affected by salt, and promoting ATPase synthesis in wheat leaves [37,38].

Plants use a variety of antioxidant techniques to decrease harmful chemicals via antioxidant enzymes. Thus, strengthening a plant's antioxidant defense can raise its tolerance to a variety of stressors. In particular, to combat the negative impact of salt stress on the permeability of the cell membrane, the most effective antioxidant enzymes, minimize lipid peroxidation in addition to guarding against oxidative harm when soil salinity stress is present. Additionally, in order to counteract the harmful effects of reactive oxygen species,

plants need to have enzymatic antioxidant activity [7,39–41]. According to [2], wheat treated with *B. phytofirmans* strain PsJN could develop more quickly and produce more enzymatic antioxidants when exposed to abiotic stressors. Furthermore, [42] proposed that *A. lipoferum*-inoculated wheat plants might be able to defend themselves against the negative impacts of abiotic stressors by altering their antioxidant defense system.

Wheat plants grown in salt-affected soil are impacted by increased root zone salt concentration, which damages plant cells due to a buildup of $Na^+$ ions in the leaf cells [43]. As a result, soil solution with a high concentration of $Na^+$ and $Cl^-$ hinders the absorption of $K^+$ and $Ca^{2+}$ and causes nutritional disruption [44,45]. According to [41], who presented findings that are in accordance with our own, adding soil with the proper PGPR strains is a crucial step in boosting nutrient uptake in maize plants grown in saline soils. In addition, PGPR and CT enhanced soil ion balance by decreasing $Na^+$ in the root zone by creating IAA and exopolysaccharide, thus enhancing $K^+$ absorption via the roots to the leaves in wheat plants [5,46]. In wheat growing in saline soils, the combined action of applied EM and CT can thereby increase $K^+$ and decrease $Na^+$ compared to the single application.

By increasing the amount and availability of soluble organic matter and soil nutrients, microbial activity increases soil fertility and is crucial for the breakdown of organic matter in soil [47,48]. Additionally, it could be related to improved soil chemistry, management of osmosis, and root secretions that raise the microbial respiration rate and increase the water-holding capacity [49]. These results are in agreement with those of several other researchers. For example, [50] found that sodic–saline soils supplemented with PGPR + biochar considerably increased soil enzyme activities (dehydrogenase and urease) compared to the control. The physicochemical characteristics and enzymatic activity of the soil were enhanced in the soil treated with PGPR in conjunction with foliar CT spraying, according to research by Omara [7], which in turn enhanced the growth of rice at various irrigation intervals.

A drop in productivity-related indicators, including the yield of grain and straw and the biological and harvest index, is caused by salt stress, which also negatively affects cell division and photosynthetic rates [51]. Similar findings were reported by [52], who claimed that the PGPR strain (*Enterobacter cloacae* ZNP-3) supports wheat plant stress tolerance by enhancing plant physiology, biochemistry, plant growth, and agronomic yield parameters. According to [53], PGPR (*Azospirillum brasilense* + *Azospirillum zeae*)-inoculated wheat plants had up to 18% higher overall grain production than un-inoculated plants. According to [54], PGPR and its combination with foliar salicylic acid application led to increases in grain production, number of spikes per square meter, and 1000-grain weight when compared to the non-treated control treatment.

## 5. Conclusions

Utilizing ecologically friendly methods to encourage the growth and production of wheat in saltine soils is urgently important as well as addressing the significant decline in the soil health and biomass. This study concluded that wheat cv. Sids 12 is a salt-sensitive plant that needs new strategies to deal with such types of cultivation conditions. Results from the two growing seasons confirmed our hypothesis that, when compared to untreated (control) plants and the application of EM or CT alone, the combined application of EM + CT significantly ($p \leq 0.05$) increased growth, yield, and nutrient uptake ($K^+$, $Na^+$, and $K^+/Na^+$ ratio) as well as photosynthetic properties, antioxidant enzymes, and soil bioactivity. Therefore, it could be suggested that the production of the wheat plant cultivated in salt-affected soil using the combined application of EM and CT is a fruitful strategy with immense benefits such as cost-effectivity and eco-friendliness, particularly in arid and semi-arid regions.

**Funding:** This research received no external funding.

**Institutional Review Board Statement:** Not applicable.

**Informed Consent Statement:** Not applicable.

**Data Availability Statement:** The data are available upon request.

**Acknowledgments:** The authors extend their appreciation to the Deanship of Scientific Research at King Khalid University for funding this work through the Large Groups Project under grant number L.G.P. 2/138/43.

**Conflicts of Interest:** The authors declare no conflict of interest.

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
