# Peer review of "Reducing the Adverse Effects of Salt Stress by Utilizing Compost Tea and Effective Microorganisms to Enhance the Growth and Yield of Wheat (Triticum aestivum L.) Plants"

_agronomy, doi:10.3390/agronomy13030823_

Round 1

Reviewer 1 Report

I think the paper is interesting, the improvement of saline soils with the use of tea compost and microorganisms. I have made some corrections to your manuscript.

In the title, at least, I would include the scientific name of the crop (wheat).

The keywords should not be repeated with the terms used in the title of the article, modify "effective microorganisms, Compost tea, ...".change the repeated words for other words.

The references, I believe that they should not be in bold, eliminate the bold in all the references of the article.

Introduction: I think the introduction is too brief, you should go more in depth on the characteristics of saline soils, the use of compost in the soil, benefits of microorganisms, in particular the ones you are going to use in the experiment.

Material and Methods:

Description of the site: This section would be much improved if the author would answer all these questions:

What type of soil has been cultivated, classify with FAO or with USDA the soil studied?

How large were the total crop fields?

What is the precipitation and temperature in the study area?

What irrigation conditions/doses did the crop have?

What methodology has been used to perform the different soil analyses?

Table 1: change the table header: Different properties (physical and biochemical) of the soil used.

Results.

Your results would be greatly improved if you include in your paper the nutrient contents (N, P, Zn, Mn, Fe, Cu) in the plant samples analyzed. Present the results and comment your data in the discussion section.

Discussion:

Lines 271-277: The beginning of the discussion should be part of the introduction of this paper.

Line 285: "Previous research"...Include bibliographic references.

Lines 303-304. That statement needs to be supported by a bibliographic reference.

Conclusions: They are fine, but I see them as too brief. Rewrite this section to include a few more paragraphs

I wish you all the best for you.

Reviewer 2 Report

1. The analytical method used in this manuscript is simple, thus further analysis is suggested to reveal the reason why the combined application (EM+CT) promotes wheat growth and yield.

2. Line 26-60. Please add relevant contents of composite tea (CT) and effective micro-organisms.

3. Line 72-78. Why these three bacterial strains were selected in the treatment of Effective Microorganisms (EM)?

4. Line 102-104. Does the different amount of nitrogen fertilizer applied between EM treatment and other treatments affect the subsequent results?

5. Line 127-131. Dose all data subject to normality and homogeneity before performing one-way ANOVA? In addition, the multiple comparison method used here is Duncan’s test instead of Turkey. The written of P value is not standard.

6. Line 132. The manuscript only focused on the difference of various parameters among different treatments. The comparative analysis between different varieties or years were lacked.

7. The "**" symbol in all tables should indicate the meaning.

8. The results showed that there was significant interaction between Cultivar and Treatment, which should be explained in detail in the discussion parts.

9. The discussion part should be according to the results section with in-depth discussion.

10. The format of references is not uniform, please check carefully.

Round 2

Reviewer 1 Report

Dear Author, 

I consider that you have taken my recommendations into account, or at least those that it has been possible.

I wish you all the best. 
